# The consumption of alcohol by adolescent schoolchildren: Differences in the triadic relationship pattern between rural and urban environments

Pilar Marqués-Sánchez[1], Enedina Quiroga Sánchez[1]*, Cristina Liébana-Presa[1], Elena Fernández-Martínez[1], Isaías García-Rodríguez[2], José Alberto Benítez-Andrades[3]

1 SALBIS Research Group, Faculty of Health Sciences, University of León, Ponferrada, Spain,
2 SECOMUCI Research Group, Escuela de Ingenierías Industrial e Informática, Universidad de León, León, Spain, 3 SALBIS Research Group, Department of Electric, Systems and Automatics Engineering, University of León, León, Spain

* equis@unileon.es

**Data Availability Statement:** All questionnaires are available at zenodo public repository, doi: https://

## Abstract

### Objective

Excessive alcohol consumption in adolescents is one of the most significant public health problems currently facing society. Social and geographical contexts contribute to the development of alcohol-related behavior in adolescents. The aim of this research is to analyze the social pattern related to alcohol consumption in adolescents based on their geographical environment.

### Methodology

We designed a descriptive cross-sectional study based on social network analysis. We recruited 196 high school students between 16 and 18 years of age to participate in the study. The methodology applied was social network analysis by means of transitivity and homophily social triads. The data were analyzed using STATA statistical software.

### Results and conclusions

A total of 58.48% of rural adolescents consumed alcohol compared to 49.52% of urban adolescents. These results demonstrate that adolescents who live in rural areas exhibit a greater risk of drinking alcohol than those who live in urban areas. The presence of transitive triads increases the probability of sharing sociodemographic attributes in such a way that it may be considered one of the causes of homophily, contributing to adolescents taking greater risks, such as consuming alcohol.

doi.org/10.5281/zenodo.3952914. All data analyzed are available at zenodo public repository, doi: https://doi.org/10.5281/zenodo.3968264.

**Funding:** The author(s) received no specific funding for this work.

**Competing interests:** The authors have declared that no competing interests exist.

## 1. Introduction

The consumption of alcohol constitutes one of the most significant problems in public health [1]. In Europe, 57% of adolescents have consumed alcohol in the past month [2]. In the USA, the data are even more alarming as 11.5% of adolescents aged between 12 and 17 have consumed alcohol to excess (a value established by the World Health Organization (WHO) in which alcohol is defined as having a very harmful effect on health) [2]. Young people are beginning to consume alcohol at a younger age, which is linked to having fun and especially with friends [3].

The social normalization that the consumption of alcohol has in our society means that adolescents do not perceive alcohol as a drug with repercussions in the short or long term and underestimate the physical, psychological, emotional and social consequences brought about by consumption both at a personal level and for society as a whole. Adolescents regard alcohol as normal and not very dangerous [4, 5]. In this sense, some studies have collated how relationships based on a group of friends play a significant role in the consumption of alcohol [4–6]. The peer group constitutes one of the most significant forms of socializing, and its influence gives rise to both voluntary and involuntary stimulation, which leads to the development of different behavioral traits [7, 8]. Consequently, for the adolescent, the consumption of alcohol is a vehicle for socializing and acceptance within the group. This fact is significant given that the adolescent will imitate peers who are socially significant to him or her [7].

The analysis of these diverse factors and the patterns of alcohol consumption have been a starting point in numerous studies. The analysis of the main factors that are associated with increased consumption of alcohol in adolescents have focused on sociodemographic factors, such as the behaviors, values and norms learned from the family and school environment. Adolescents view alcohol as one of the positive aspects in their social lives [9]. International studies have established how adolescents who live in rural areas have a greater propensity to consume alcohol compared with those who live in urban areas [10]. Historically, it has been thought that isolation in rural areas gave rise to a delay in the consumption of any type of drug. However, research shows that the consumption of alcohol is similar between both groups or even greater in a rural environment. However, there are a series of qualitative differences that highlight that the consumption of alcohol by both urban and rural adolescents shows signs of their own identity [10].

Based on the aforementioned information, it is fundamental that we not only place importance on the consumption of alcohol itself but also on the factors that promote it. Diverse sociological research has been performed on the homogeneity of groups based on homophily in different environments [11–13]. Homophily is a concept that means that people tend to have more links with others who have similar social characteristics [11–13]. In this way, homophily may be evidenced as a fundamental mechanism in the constitution of social groups. The adolescent together with his or her friends comprises a social group with their own norms and values, which allows them to satisfy their needs for integration and acceptance [4].

The study of homophily has undergone a significant advance through the theory of social networks [14]. The theory of social networks analyzes the ties that link members of society. Thus, the theory of social networks adapts itself both theoretically and analytically to the concept of homophily as actors feel attracted to forming ties with others who are similar to themselves independently of their structural organization [15, 16].

Network analysts have debated homophily using the language of the Theory of Balance and Transitivity, basic elements that reflect how interpersonal relationships tend towards a state of equilibrium [17]. The motivation for speaking of homophily and the Theory of Balance and Transitivity is relevant because the adolescent is seeking integration into the events

surrounding him or her while trying to maintain a constant equilibrium. Empirical studies have confirmed that the principle of equilibrium and transitivity are applied in approximately 70 to 80% of small groups, thus making them the most favorable scenario for a network to remain stable [16, 18].

For many authors, the structure of the relationship in the network begins with triads because they constitute the beginnings of a "society" [19]. A triad is a subset of the social network composed of three actors and the possible ties between them [20]. A transitive triad is one that is established between the nodes with the following sequence: If X connects with Y, and Y connects with Z, then X connects with Z. In contrast, a nontransitive triad involve a greater number of possibilities [20]. The number of people who comprise a network, the number of links that each individual has with the others, the position that each individual occupies within the network, its composition, and the relationships that are established between the members of the network are extremely relevant in revealing the behavior and the environment in which they propagate [21].

From the point of view of Simmel (1950), in the triad, the third individual may be an impartial environment and a mediator, reading "*el tertius gaudens*", that is, "the third one that benefits". The third individual may side up with one of the other two individuals and in this way achieve his or her own interests or act as an intermediary to obtain the necessary benefit.

The method used to analyze these data is social network analysis (SNA), and the behavior patterns are specifically analyzed and interpreted [22]. SNA is an emerging research perspective within the social and behavioral spheres that utilizes a series of methods, models and applications expressed in terms of relational concepts, which allows us to determine the types of relationships that are established between adolescents and their peers and how these relationships might influence the development of their habits [23–25].

Based on the aforementioned information, we have become aware that there is a lack of studies that analyze in great detail the type of connections among adolescents and how these connections generate a balance in the small groups that influence and foster the consumption of alcohol. Thus, the objective of this study was to analyze (i) the triad pattern and (ii) the possible guidelines according to the geographical environment of the adolescent. This study seeks to demonstrate that (1) there are criteria for homophily and (2) individuals who possess the same social attributes are more likely to stay in contact with each other and therefore develop similar practices that may include the consumption of alcohol.

The application of the triads within the SNA and their link with the consumption of alcohol among adolescents is useful in investigating the reasons for grouping among the actors and thus research their influence on the formation of subgroups.

## 2. Materials and methods

### Participants

The total population of the different institutes was 270 adolescents. Of these, 195 adolescents or their parents in the case of adolescents under 18 years of age signed the informed consent form.

The sample was composed of 195 adolescents aged between 16 and 19 attending rural and urban public schools in the area of El Bierzo (León, Spain) during the 2015–2016 academic year. The selection was made with the consent of the project from the centers. We also took into account the relationships of the research team with the directors and professors, which facilitated the launch of the research. A search for educational centers in rural and urban settings was performed to obtain a sample of similar size that would allow us to compare both rural and urban areas.

There are various definitions for characterizing rural settings. The Ministry of the Environment defines the proportion and density of the population as the most commonly used criterion. In Spain, rural areas are defined as "A geographic space comprising the aggregation of municipalities or smaller local entities with a population <30,000 inhabitants and a density <100 inhabitants per km$^2$" [26]. Those urban areas with a population of greater than 50,000 are considered urban areas [26, 27]. It should be noted that as the rural areas were more isolated, inhabitants had less access to various drugs [28, 29]. Study recruitment was incidental; that is, participation in the study was voluntary without bias of any kind since not having the informed consent signed by the parents was considered a criterion for exclusion. From the total sample, 53.85% of the students (n = 105) were women, and 46.15% (n = 90) were men. The average age of the group was 17 years (SD: 0.82). Regarding the rural or urban character of the schools participating, 51.28% (n = 100) of the total number of participating students studied at rural schools, whereas 48.72% (n = 95) studied in an urban environment.

### Ethical approval

The study was approved by the University Ethics Committee (ETICA-ULE-003-2015).

### Instruments

A questionnaire was sent online by means of validated instruments (AUDIT test questionnaire) and other "ad hoc" methods to cover the interest in the research [30, 31]. To collect data related to the adolescent's consumption of alcohol, the AUDIT test questionnaire was applied [32, 33]. The complete questionnaire used is available at https://doi.org/10.5281/zenodo.3965836.

To connect relational-type data, a question was formulated with respect to the network of friends and consumption drawn up in accordance with revised literature with the SNA methodology. The question related to the network of friends was formulated for each student as follows: "Using the list below, indicate how much time you spend with your classmates" [34, 35]. Thus, the adolescents evaluated the intensity of their contact. Equally, students indicated which of their colleagues consumed alcohol with them when they went out based on the following statement: "Highlight on the following list those colleagues in class with whom you go out and have a drink with". Thus, it was possible to determine dichotomously the existence of a relationship within a network of contacts for the adolescent's consumption of alcohol.

### Procedure

After receiving the approval of the Bioethical Committee of the University of Leon, contact was made with the academic head of each school to explain the research and request informed consent. The research team coordinated with the teachers. All of the data collected were stored in the electronic file and automated using a method especially created for this research. The data on the identification of each student were encoded instantaneously in the name of the file by means of an online questionnaire platform with the aim of maintaining anonymity in compliance with the Law on the Protection of Personal Data (Constitutional Law 3/2018, on 5 December on the Protection of Personal Data and Guaranteeing digital rights).

### Data analysis

To analyze the data, the STATA 14.0 program (StataCorp LLC, 4905 Lakeway Drive, College Station, TX 77845 USA) was used for statistical data, whereas relational data were assessed using the UCINET 6.649 Program [36]. All of the parameters were analyzed based on a

confidence interval of 95%, and p<0.05 indicated significant values. The G*Power 3.1.9.6 program was used to perform the power analysis. To determine the correlations, we used Pearson's correlation coefficient. Age was used as a control variable that had values between 16 and 19 years old.

Regarding the relational character, an initial matrix was established in which, by means of responses assigned to the time frequencies (1: we never coincided, 5: we are always together), three matrices of different proximities were obtained (intensity of minimum, intermediate and maximum contact) on the basis of three criteria of dichotomization (Table 1) [37]. After that, an analysis of the triads together with a transformation of the friendship networks was performed in accordance with the following compositions: a) minimum contact as this signifies the start of a relationship, b) gender, and c) contact with that network for consumption. The establishment of triads in accordance with minimum contact, gender, and contact network for consumption was motivated by the perspective of the concept of homophily and following the recommendations of McPherson et al. (2001) and Louch (2000) [15, 38]. The census of transitive and intransitive triads according to Davis was analyzed [16]. All data are available at https://doi.org/10.5281/zenodo.3968264.

## 3. Results

The objective of establishing the existing triads will allow us to know why and how the operation of the complete network works in a group.

### Sociodemographic results

We analyzed the consumption of excessive alcohol in the samples studied. It was confirmed that 110 students (56.41%) presented a no risk of consumption compared to 85 students (43.59%) who presented a risk of consumption. The risk of alcohol consumption was significantly associated with female gender. Specifically, we observed that 50.48% of women exhibit a risk of excessive alcohol consumption compared to 49.52% of men who were at risk of excessive alcohol consumption $X^2$ (2, N = 105) = 4.40, p = 0.036. The average age of the sample for alcohol consumption was 13.4 years old.

Regarding geographical location, a statistically significant association is observed in which participants living in the rural environment (53.64%) exhibit a greater rate of alcohol consumption compared with those in the urban environment (46.36%). This finding explains why adolescents who live in rural environments exhibit increased consumption of alcohol $X^2$ (2, N = 100) = 4.50, p = 0.04.

### Triad pattern

The calculation of all types of triads, including transitive and intransitive, was performed in each classroom with respect to gender, contact network and contact network of minimum intensity. The most representative data are presented in Table 2.

**Table 1. Dichotomizations based on the intensity of contact.**

| Contact Intensity | Values without contact | Values with contact |
|---|---|---|
| Minimum contact | 1 | 2, 3, 4 & 5 |
| Intermediate contact | 1 & 2 | 3, 4 & 5 |
| Maximum contact | 1, 2 & 3 | 4, 5 |

Source: Prepared by authors based on data from Arias (2017).

**Table 2. Dichotomizations based on the intensity of contact.**

|  | RURAL | URBAN |
|---|---|---|
| **Gender Composition** | Triad 102 (Intransitive) | Triad 102 (Intransitive) |
|  | 68.00% | 75.25% |
|  | Triad 300 (Transitive) | Triad 300 (Transitive) |
|  | 26.83% | 25.50% |
| **Contact network** | Triad 12 (Intransitive) | Triad 12 (Intransitive) |
|  | 15.33% | 29.75% |
|  | Triad 102 (Intransitive) | Triad 102 (Intransitive) |
|  | 10.12% | 18.00% |
| **Minimum intensity** | Triad 102 (Intransitive) | Triad 12 (Intransitive) |
|  | 14.50% | 21.75% |
|  | Triad 111U (Intransitive) | Triad 102 (Intransitive) |
|  | 12.83% | 18.5% |

Transitive triads exhibit a type of balance in which if X directs a loop to Y and Y directs a loop to Z, then X also directs a loop to Z. Intransitive triads do not exhibit any type of balance between actors.

There are four most representative triads in this study according to the variables studied: T012, T102, T111U and T300. Fig 1 presents the representation and explanation of these four types of triads according to Wasserman.

Transient triads are a more favorable scenario than a stable network [19]. Based on this notion, the urban environment presents a greater number of transitive triads in the minimum intensity contact networks (1084/10887) and in the composition based on gender (2957/11415); however, in relation to consumption, the number of transitive triads is greater within the rural environment (913/4506).

In the correlation analysis between the transitive triads (which demonstrate a balance) of the contact networks for consumption and the triads of the minimum intensity networks and

| Triad 012 (Intransitive) | Triad 102 (Intransitive) | Triad 111U (Intransitive) | Triad 300 (Transitive) |
|---|---|---|---|

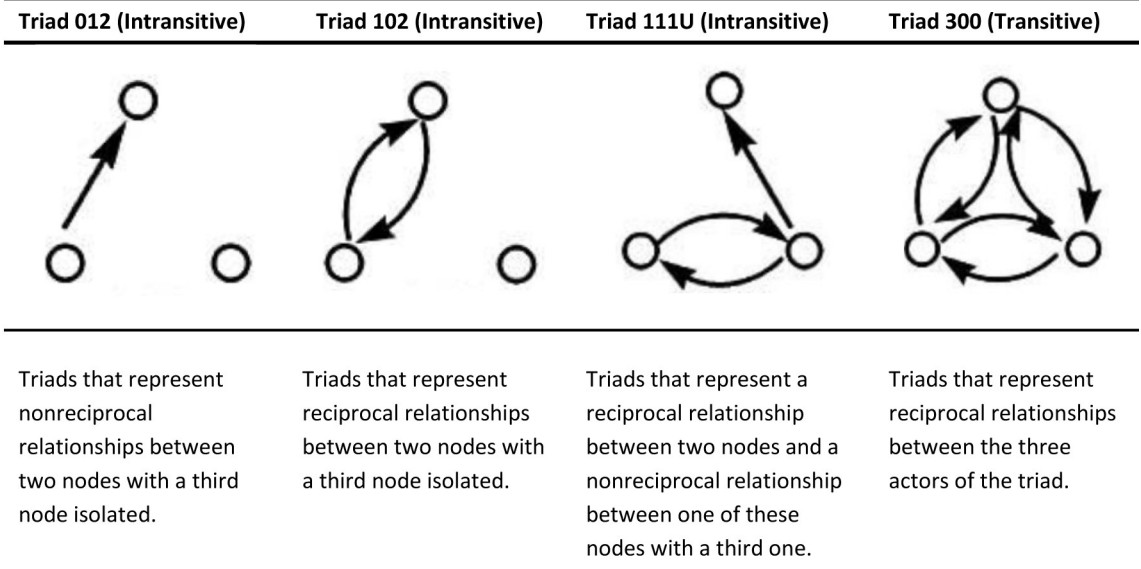

| Triads that represent nonreciprocal relationships between two nodes with a third node isolated. | Triads that represent reciprocal relationships between two nodes with a third node isolated. | Triads that represent a reciprocal relationship between two nodes and a nonreciprocal relationship between one of these nodes with a third one. | Triads that represent reciprocal relationships between the three actors of the triad. |
|---|---|---|---|

**Fig 1. Description of the four most representative triads in this research (T012, T102, T111U and T300).**

the friendship network per gender composition (Table 3), we established that the urban environment networks are related in this state of balance. The statistical power for the tests is good in the majority of cases.

## 4. Discussion and conclusions

The purpose of this study was to analyze the structural pattern of triad relationships in both rural and urban networks. These findings were complimented with other prevailing data given that the methodology emerging from SNA has been conjugated with the most common data focusing on epidemiology.

Our findings with respect to the prevalence of consumption confirm that 86.7% of adolescents had consumed alcohol once in their life. In addition, 43.5% of them drank to excess, while only 13.3% of the adolescents had never tried alcohol. Regarding gender, 62.4% of the female population presented an excessive consumption of alcohol compared to 37.7% of the masculine gender. The average age of first consumption was 13.2 (SD = 1.68) years old for males and 13.6 (SD = 1.34) years old for females. These results are consistent with other similar studies in Europe, such as the European Monitoring Centre for Drugs and Drug Addiction (EMCDD, 2019), which reported that alcohol is the substance most often consumed by adolescents. Regarding gender and the excessive consumption of alcohol, our study demonstrates that females exhibit a greater propensity for behavior leading to the excessive consumption of alcohol compared with males (62.4 vs. 37.6%; $\chi^2$ = 4.4; p = 0.04). Our findings are consistent with other studies, which confirm a greater percentage of excessive consumption in females [39, 40]. The actual increase in the incidence of alcohol consumption in women may be justified as peer pressure that females experience. The learning of these roles begins in childhood and continues into adolescence and adulthood as the woman tries to insert herself into a society that is still traditional in many aspects [39].

Regarding the consumption of alcohol and its relationship with rural or urban residence, the results of our study revealed increased consumption in the rural environment, which is consistent with studies that have confirmed that young people who live in rural areas exhibit an increased probability of excessively consuming alcohol compared with those who live in urban areas [10]. It seems that the context of a rural environment could be related to the consumption of alcohol based on the home education of adolescents [10]. Specifically, they live in a situation with less parental control and therefore are used to drinking alcohol both more frequently and in larger amounts.

**Table 3. Logistic regression results.** Transitive triadic correlation of the contact network for consumption with the contact network by minimum intensity and gender composition.

| Geographical environment | Contact networks triads for consumption | Current contact network correlation (minimal intensity) | | Gender composition correlation | |
|---|---|---|---|---|---|
| | | Pearson Correlation | Power | Pearson Correlation | Power |
| **Rural** | **CLASSROOM_1A** | -0.05 | 0.11 | -0.25 | 0.94 |
| | **CLASSROOM _2A** | 0.69 | >0.99 | -0.17 | 0.66 |
| | **CLASSROOM_1B** | -0.11 | 0.34 | -0.12 | 0.39 |
| | **CLASSROOM_2B** | 0.40 | >0.99 | -0.30 | 0.99 |
| | **CLASSROOM_1C** | 0.83 | >0.99 | 0.49 | >0.99 |
| | **CLASSROOM_2C** | 0.69 | >0.99 | 0.80 | >0.99 |
| **Urban** | **CLASSROOM_1AH** | 0.62 | >0.99 | 0.99 | >0.99 |
| | **CLASSROOM_1BC** | 0.77 | >0.99 | 0.99 | >0.99 |
| | **CLASSROOM_2AH** | 0.97 | >0.99 | 0.68 | >0.99 |
| | **CLASSROOM_2BC** | 0.89 | >0.99 | 0.99 | >0.99 |

In relation to the triad pattern, the studies reported in the literature are simple given that the application of transitivity to the analysis of small groups has proven to be an obstacle in its use in sociological research [38]. The theories of triadic transitivity have largely not been applied to the relationship between the structure and content of the relationships. Thus, very few studies have asked whether the triadic transitivity in the networks is affected by the attributes of the members of the network [38]. These types of questions were of interest in this study given that the analysis of the triadic structures might contribute to knowledge on the overall attributes shared by the network [22], such as gender, friendship or consumption.

The results obtained in this work demonstrated that the structural pattern in an urban environment is defined by means of a large number of transitive and complete triads in the minimum intensity contact networks with regard to gender. In other words, it could be said that urban adolescents who wish to consume alcohol exhibit an increased density in their network (measured by gender). Within this density, there are many subsets composed of three actors who have a very strong link to each other.

This aspect is not visible in the pattern of rural adolescents. In interpreting these conclusions, we have paid special attention to homophily and transitivity. The cohesion of the network is considered by studying the number of existing transitional triads and linking this cohesion to the concept of homophily. Thus, it is certified that homophily shapes the existing triads between contacts. In our study, this information allows us to interpret that people with a homophilic relationship share common characteristics, which makes communication and the formation of a relationship much easier.

Continuing along these lines, the work within the SNA that showed the link between the concept of homophily and consumption of alcohol is diverse, defending the criteria of homophily as the basis for the origin of stable networks for consumption [41–43] Regarding geographical differences, the literature reflects that adolescents choose to have relationships with those who have similar social characteristics [15]. In this aspect, the urban environment allows a greater distribution with regard to gender such that adolescents are able to choose their friendships by looking for similarities. However, the rural environment, is characterized by social dependence. The lower population density means that adolescents cannot resort to similarities to choose their friendships and exclusively rely those found in the population with which they interact [41, 44]. The greater presence of complete triads in the urban environment within our study represents nodes that are completely connected to each other. This finding indicates a greater intensity of contacts.

In relation to this aspect, Moody and White (2003) have noted the importance of the presence of complete triads in the origin of networks with better conditions for the dissemination of information [45] or actions [46]. These complete triads relate the fact that the network is highly connected between itself and with aspects of homophily such that this cohesion would permit common expectations or behaviors to be shared. Alshamsi et al. (2015) posited that with regard to the high density of the population, cities have many advantages, including better and easier social interaction and exchange of information [47]. Their results reveal the existence of the hemophiliac community at the urban level with implications on the creation of well-being and the repercussions of an unhealthy lifestyle behaviors, such as the consumption of alcohol.

This question lends support to the facts stated in the results that the significant presence of transitive triads increases the probability of sharing sociodemographic attributes such that one could consider the presence of transitive triads as one of the causes of homophily. Continuing along these lines, the studies have revealed diverse links between the levels of homophily with attributes, such as gender, age, and behavior, such as the consumption of alcohol [48].

The information derived from this discussion may be useful for studying the relationships between adolescence and the propagation and acquisition of the habit of consuming alcohol given that it even deepens the microstructures, such as the triads within the network, in the analysis of the social influence as part of the process of the contagion of risky behavior [49, 50]. Having precise knowledge at the macro level of who is in contact with people who consume alcohol allows us to better understand the relational process at the macro level. This relational process can involve a school, a population, etc. Therefore, the triadic approach in the study of the creation of homophilic subgroups among adolescents and the study of bonding attributes seems to be suitable for addressing the association process between individuals. Finally, the fact that we are able to know how adolescents associate may offer us positive results of a preventative nature.

The SNA is a useful tool for understanding the social patterns of adolescents. Knowledge is critical in determining how the network would facilitate the planning of multifactorial strategies in the environment.

As limitations to the research, we note that this type of transversal study does not allow for the establishment of causal relationships in the results. In addition, the size of the sample is a limitation when generalizing the results. Another limitation is the method of the self-questionnaire used given that we must assume that the individual is telling the truth.

## Supporting information

**S1 File.**
(TXT)

**S2 File.**
(TXT)

## Author Contributions

**Conceptualization:** Pilar Marqués-Sánchez, Enedina Quiroga Sánchez, Elena Fernández-Martínez, Isaías García-Rodríguez.

**Data curation:** Cristina Liébana-Presa, José Alberto Benítez-Andrades.

**Formal analysis:** Pilar Marqués-Sánchez, Enedina Quiroga Sánchez, Elena Fernández-Martínez.

**Investigation:** Pilar Marqués-Sánchez, Enedina Quiroga Sánchez, José Alberto Benítez-Andrades.

**Methodology:** Enedina Quiroga Sánchez, Cristina Liébana-Presa, Elena Fernández-Martínez.

**Project administration:** José Alberto Benítez-Andrades.

**Resources:** Cristina Liébana-Presa, José Alberto Benítez-Andrades.

**Software:** Isaías García-Rodríguez, José Alberto Benítez-Andrades.

**Supervision:** Isaías García-Rodríguez, José Alberto Benítez-Andrades.

**Validation:** Enedina Quiroga Sánchez, Cristina Liébana-Presa, Elena Fernández-Martínez, Isaías García-Rodríguez.

**Visualization:** Cristina Liébana-Presa.

**Writing – original draft:** Pilar Marqués-Sánchez, Enedina Quiroga Sánchez.

**Writing – review & editing:** Pilar Marqués-Sánchez, Enedina Quiroga Sánchez, Isaías García-Rodríguez, José Alberto Benítez-Andrades.

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
