## [Decision Letter · Decision Letter 0]

27 Jul 2020

PONE-D-20-13204

The consumption of alcohol by adolescent schoolchildren: Differences in the triadic relationship pattern between rural and urban environments.

PLOS ONE

Dear Dr. Quiroga Sánchez,

Thank you for submitting your manuscript to PLOS ONE. After careful consideration, we feel that it has merit but does not fully meet PLOS ONE’s publication criteria as it currently stands. Therefore, we invite you to submit a revised version of the manuscript that addresses the points raised during the review process.

We look forward to receiving your revised manuscript.

Kind regards,

Joel Msafiri Francis, MD, MS, PhD

Academic Editor

PLOS ONE

Journal Requirements:

4. Your ethics statement must appear in the Methods section of your manuscript. If your ethics statement is written in any section besides the Methods, please move it to the Methods section and delete it from any other section. Please also ensure that your ethics statement is included in your manuscript, as the ethics section of your online submission will not be published alongside your manuscript.

Reviewers' comments:

Reviewer's Responses to Questions

**Comments to the Author**

1. Is the manuscript technically sound, and do the data support the conclusions?

Reviewer #1: Yes

Reviewer #2: Partly

2. Has the statistical analysis been performed appropriately and rigorously? 

Reviewer #1: I Don't Know

Reviewer #2: No

3. Have the authors made all data underlying the findings in their manuscript fully available?

Reviewer #1: No

Reviewer #2: Yes

4. Is the manuscript presented in an intelligible fashion and written in standard English?

Reviewer #1: Yes

Reviewer #2: No

5. Review Comments to the Author

Reviewer #1: Recommendation: Minor Revisions

Thank you for the opportunity to review for PLoS One. Thank you to the authors for their hard work and interesting study. Overall, there might be a relevant and important paper here, however some sections are unclear and need refining if they are to sell the novelty of the paper fully. I will note that I am not an expert on social network analysis (I only have a basic quantitative understanding of the specific methodologies involved) and the paper should be considered by a relevant statistician before a decision is made.

Strengths

- There are few empirical social network analyses out there due to the difficulty in obtaining data suggesting a potential novel study design

- Potentially interesting analyses on the network structure and how varies by some basic sociodemographic issues

- Well written discussion

Areas for Improvement

- Line 39: ‘In this sense, different research’ doesn’t quite work with the rest of the statement, suggest revise

- Line 44: Drop use of ‘extremely’ as no need to overplay the importance of the issue

- Lines 94-94: Repeated use of (i) for different points is confusing to the reader and suggests overlap when there is not. Please revise.

- The introduction is not structured effectively and jumps about a lot. It would benefit from thinking about the flow of ideas and the content being discussed. It also doesn’t feel like there is much novelty here. I am sure there is, but it is not explained clearly in particular that the premise of the study appears to suggest that we already know a lot of this.

- Line 102: What is ‘DE’, please define in first use

- Line 149: ‘We analysed the la existence’ needs correcting

- Line 153: Unsure what is the point of the statistical test here as the numbers appear to be comparing whether there is a difference between risk and no risk in women. Although a ‘statistically insignificant’ effect is reported, there is a <1% difference here which suggests in reality it isn’t important. I would have expected a comparison of the metric between women and men here instead (i.e. were women more or less at risk compared to men) - this might be what you did but it isn’t clear that it is.

- There are many results to report meaning the section is rather brief. I think this is more an issue in how they are discussed and some of their novelties are not teased out. It might need greater clarity especially to help guide a lay reader which will be important to increase the awareness and reach of your paper

- No discussion of limitations of research in the discussion. What were the major weaknesses or opportunities for future research? How representative were your data? How might you improve on your study? What data could you have collected to make it stronger? We would normally expect some reflection over the strengths and weaknesses of a paper.

- Data and analytical code are not openly available. If possible (more for the former than the latter which should be fine), this should be made available to adhere to PLoS One guidance. This will also improve the impact of your paper.

Reviewer #2: Thank you for the opportunity to the manuscript entitled, The consumption of alcohol by adolescent schoolchildren: Differences in the triadic relationship pattern between rural and urban environments submitted to PLOS ONE (PONE-D-20-13204). The study examines alcohol use among a sample of adolescents to better understand the geographic contribution to alcohol consumption. The study also examines the influence of peer groups by assessing triadic relationships. The manuscript has significant grammatical errors and needs copyediting. The introduction is brief and needs additional discussion on the contribution of geographic area on alcohol availability and alcohol consumption. The methods section is most problematic; there are essential details that are missing that limit interpretation of the results. In its current form, the manuscript is not ready for publication. Specific comments are listed below.

Line 37, define risk limits.

Line 38 should be supported by additional citations. Similarly, line 47 should be supported by additional citations. The authors should review the literature on neighborhood environment (and rural/urban) and alcohol availability and consumption.

The language is too causal in some areas “it is important that nowadays”

Line 57 should be rewritten

Line 89 should be rewritten

The last paragraph is vague. The authors should be specific when discussing the aims (e.g. including the outcome measure).

Are state schools similar to public schools in the US? How was rural and urban defined? Differences in alcohol availability?

Were adolescents under 18 able to consent? Was there parental consent?

What does “The recruitment was incidental” mean?

Response rate? Compensation for completing the assessment?

“ad-hoc” ones is too casual. This sentence needs a citation. Which validated instruments were used?

How many questions were included? How was the outcome measure defined?

How were schools and areas selected? The year(s) of data collection should be included.

Inclusion and exclusion criteria?

The data analysis section should be rewritten. Alpha should be less than 0.05, not less than or equal to. Any control variables? What statistical test was used? Any missing data? Power analysis?

There are areas in the results that are not English “la existence” and “consumption de alcohol.”

Line 160 belongs in the methods

Table 4 does not stand alone and is difficult to interpret.

The results cannot be fully evaluated without the missing information in the methods section.

6. PLOS authors have the option to publish the peer review history of their article (what does this mean?). If published, this will include your full peer review and any attached files.

Reviewer #1: No

Reviewer #2: No

---

## [Author Response · Author response to Decision Letter 0]

31 Jul 2020

First, we would like to thank the editorial committee and reviewers for their work on our manuscript. We then answered each of the questions and suggestions from the two reviewers.

Reviewer 1

Reviewer #1: Recommendation: Minor Revisions

Thank you for the opportunity to review for PLoS One. Thank you to the authors for their hard work and interesting study. Overall, there might be a relevant and important paper here, however some sections are unclear and need refining if they are to sell the novelty of the paper fully. I will note that I am not an expert on social network analysis (I only have a basic quantitative understanding of the specific methodologies involved) and the paper should be considered by a relevant statistician before a decision is made.

Strengths

- There are few empirical social network analyses out there due to the difficulty in obtaining data suggesting a potential novel study design

- Potentially interesting analyses on the network structure and how varies by some basic sociodemographic issues

- Well written discussion 

Answer 1.0:

We thank reviewer one for his appreciation of the research. We are very happy to know that you find our research interesting. We hope to be able to satisfy all the proposals and comments you indicate in this review.

Areas for Improvement

Q1.1.: Line 39: ‘In this sense, different research’ doesn’t quite work with the rest of the statement, suggest revise

A1.1.: We have changed “different research” to “some studies”. 

Q1.2.: Line 44: Drop use of ‘extremely’ as no need to overplay the importance of the issue

A1.2.: It has been removed.

Q1.3.: Lines 94-94: Repeated use of (i) for different points is confusing to the reader and suggests overlap when there is not. Please revise.

A1.3.: It has been changed to (1) and (2) in the second phrase. We hope it's not confusing anymore.

Q1.4.: The introduction is not structured effectively and jumps about a lot. It would benefit from thinking about the flow of ideas and the content being discussed. It also doesn’t feel like there is much novelty here. I am sure there is, but it is not explained clearly in particular that the premise of the study appears to suggest that we already know a lot of this.

A1.4.: We have modified the introduction. We hope that after the proposed changes that have been made in this section, we will have fulfilled this requirement as well.

Q1.5.: Line 102: What is ‘DE’, please define in first use

A1.5.: Sorry, DE is the English translation of SD (Standard Deviation). It has been changed.

Q1.6.: Line 149: ‘We analysed the la existence’ needs correcting

A1.6.: It has been corrected.

Q1.7.: Line 153: Unsure what is the point of the statistical test here as the numbers appear to be comparing whether there is a difference between risk and no risk in women. Although a ‘statistically insignificant’ effect is reported, there is a <1% difference here which suggests in reality it isn’t important. I would have expected a comparison of the metric between women and men here instead (i.e. were women more or less at risk compared to men) - this might be what you did but it isn’t clear that it is.

A1.7.: We have modified the manuscript with the following sentence:

“The risk consumption of alcohol is significantly seen to be associated with the female gender, in which we observed that 50.48% of women has a risk alcohol consumption compared to 49.52% of men who were at risk of alcohol consumption (50.48 Vs 49.52%; X2=4.40; p=0.036).”

Q1.8.: There are many results to report meaning the section is rather brief. I think this is more an issue in how they are discussed and some of their novelties are not teased out. It might need greater clarity especially to help guide a lay reader which will be important to increase the awareness and reach of your paper.

A1.8.: We have modified the discussion. We hope that after the proposed changes that have been made in this section, we will have fulfilled this requirement as well.

Q1.9.: No discussion of limitations of research in the discussion. What were the major weaknesses or opportunities for future research? How representative were your data? How might you improve on your study? What data could you have collected to make it stronger? We would normally expect some reflection over the strengths and weaknesses of a paper.

A1.9.: We have modified the discussion. We hope that after the proposed changes that have been made in this section, we will have fulfilled this requirement as well.

Q1.10.: Data and analytical code are not openly available. If possible (more for the former than the latter which should be fine), this should be made available to adhere to PLoS One guidance. This will also improve the impact of your paper.

A1.10.: Attending to this suggestion, we have uploaded the questionnaires used at zenodo https://doi.org/10.5281/zenodo.3965836 and all database is available at zenodo https://doi.org/10.5281/zenodo.3968264 .

We hope that after all the proposed changes that have been made, we will have fulfilled this requirement as well.

 

Reviewer 2

Reviewer #2: Thank you for the opportunity to the manuscript entitled, The consumption of alcohol by adolescent schoolchildren: Differences in the triadic relationship pattern between rural and urban environments submitted to PLOS ONE (PONE-D-20-13204). The study examines alcohol use among a sample of adolescents to better understand the geographic contribution to alcohol consumption. The study also examines the influence of peer groups by assessing triadic relationships. The manuscript has significant grammatical errors and needs copyediting. The introduction is brief and needs additional discussion on the contribution of geographic area on alcohol availability and alcohol consumption. The methods section is most problematic; there are essential details that are missing that limit interpretation of the results. In its current form, the manuscript is not ready for publication. Specific comments are listed below.

A2.0.: We thank reviewer one for his appreciation of the research. We are very happy to know that you find our research interesting. We hope to be able to satisfy all the proposals and comments you indicate in this review.

Q2.1.: Line 37, define risk limits.

A2.1.: It has been defined in the manuscript:

“alcohol to risk limits (a value established by the WHO in which alcohol is defined as having a very harmful effect on health) [2].”

Q2.2.: Line 38 should be supported by additional citations. Similarly, line 47 should be supported by additional citations. The authors should review the literature on neighborhood environment (and rural/urban) and alcohol availability and consumption.

A2.2.: We have added more references to these paragraphs.

Q2.3.: The language is too causal in some areas “it is important that nowadays”

A2.3.: We have hired a native English translator to review the entire manuscript. We hope this version will be more readable. 

Q2.4.: Line 57 should be rewritten 

Line 89 should be rewritten 

The last paragraph is vague. The authors should be specific when discussing the aims (e.g. including the outcome measure).

A2.4.: We have rewritten lines 57, 89 and the last paragraph of the introduction. In addition, this section was modified following the suggestions of reviewer 1.

Q2.5.: Are state schools similar to public schools in the US? 

A2.5.: Yes, state schools are the same as public schools. We've changed the terminology used.

Q2.6.: How was rural and urban defined?

A2.6.: We have added this text to the manuscript:

“The difference between rural and urban is defined by the proportion of the population. Those nuclei with a population of more than 50,000 are considered to be urban areas [26,27]”.

Q2.7.: Differences in alcohol availability?

A2.7.: We have added this text to the manuscript:

“It should be noted that the rural areas being more isolated had less access to various drugs [28,29].”

Q2.8.: Were adolescents under 18 able to consent? Was there parental consent?

A2.8.: We have added this text to the manuscript:

“The initial population was 270 adolescents, 195 of whom signed the informed consent form or their parents in the case of adolescents under 18.”

Q2.9.:What does “The recruitment was incidental” mean?

A2.9.: We have added this text to the manuscript:

“that is, incorporation into the study was voluntary, without bias of any kind.”

Q2.10.: Response rate? Compensation for completing the assessment?

A2.10.: We have added this text to the manuscript:

“The initial population was 270 adolescents, 195 of whom signed the informed consent form or their parents in the case of adolescents under 18.”

Q2.11.: “ad-hoc” ones is too casual. This sentence needs a citation. Which validated instruments were used?

A2.11.: We have changed the word “ones” and we have added two references and we have changed the text to:

“, by means of validated instruments (AUDIT test questionnaire) and other “ad-hoc” ones to cover the interest of the research [27,28]”

Q2.12.: How many questions were included? How was the outcome measure defined?

A2.12.: In relation to the first question

We have added this text to the manuscript:

The complete questionnaire used is available at : https://doi.org/10.5281/zenodo.3965836

With regard to the “outcome measure”, the Alcohol Use Disorder Identification Test (AUDIT) is composed of ten questions that explore three domains: risk consumption, symptoms of dependence, and harmful consumption. Each question is scored from 0 to 4. Cherpitel [https://doi.org/10.1016/0376-8716(95)01199-4] proposes cut-off points to identify risk consumption, harmful consumption and dependence. Using the cut-off point at 8 for men and 6 for women, AUDIT is a sensitive test (51-97%) for detecting harmful alcohol use, abuse or dependence. It is currently the most recommended risk screening technique, based on its simplicity of application and its focus in the recent past (last year).

Q2.13.: How were schools and areas selected? The year(s) of data collection should be included.

A2.13.: We have added this text the manuscript:

“The selection was carried out taking into account the receptivity to the project by the centres, the relations of the research team with the directors and teachers, and the search for educational centres with which to obtain a sample of similar size, which would allow us to compare the rural and urban areas.”

And we have added this text including the year:

"… attending rural and urban state public schools in the area of El Bierzo (León, Spain), during the 2015-2016 academic year.”

Q2.14.: Inclusion and exclusion criteria?

A2.14.: We have added this text to the manuscript:

Since not having the informed consent signed by the parents was considered a criterion for exclusion.

Q2.15.: The data analysis section should be rewritten. Alpha should be less than 0.05, not less than or equal to. 

A2.15.: This was a transcription error, the criterion of p<0.05 was used. Therefore, the analysis performed was correct. We have changed it in the manuscript.

Q2.16.: Any control variables? 

A2.16.: We have added this text to the manuscript:

Age was used as a control variable, being between 16 and 19 years old.

Q2.17.: What statistical test was used? 

A2.17.: We have added this text to the manuscript:

“To determine correlations, we used Pearson’s correlation coefficient.”

Q2.18.: Any missing data? 

A2.18.: No data were lost because all participants filled in all the questions.

Q2.19.: Power analysis?

A2.19.: The power analysis has been carried out with the program G*Power 3.1.9.6 and the values obtained in table 4 have been added.

Q2.20.: There are areas in the results that are not English “la existence” and “consumption de alcohol.”

A2.20.: We have corrected these mistakes.

Q2.21.: Line 160 belongs in the methods

A2.21.: We have moved this part sentence to methods.

Q2.22.: Table 4 does not stand alone and is difficult to interpret.

A2.22.: We have modified the names of the "classrooms" that were in Spanish. We hope that now the table is more understandable.

Q2.23.: The results cannot be fully evaluated without the missing information in the methods section.

A2.23.: We hope that after all the proposed changes that have been made, we will have fulfilled this requirement as well.

---

## [Decision Letter · Decision Letter 1]

1 Sep 2020

PONE-D-20-13204R1

The consumption of alcohol by adolescent schoolchildren: Differences in the triadic relationship pattern between rural and urban environments.

PLOS ONE

Dear Dr. Quiroga Sánchez,

Thank you for submitting your manuscript to PLOS ONE. After careful consideration, we feel that it has merit but does not fully meet PLOS ONE’s publication criteria as it currently stands. Therefore, we invite you to submit a revised version of the manuscript that addresses the points raised during the review process.

We look forward to receiving your revised manuscript.

Kind regards,

Joel Msafiri Francis, MD, MS, PhD

Academic Editor

PLOS ONE

Additional Editor Comments (if provided):

Please kindly address the additional reviewer comments carefully.

Reviewers' comments:

Reviewer's Responses to Questions

**Comments to the Author**

1. If the authors have adequately addressed your comments raised in a previous round of review and you feel that this manuscript is now acceptable for publication, you may indicate that here to bypass the “Comments to the Author” section, enter your conflict of interest statement in the “Confidential to Editor” section, and submit your "Accept" recommendation.

Reviewer #1: All comments have been addressed

Reviewer #2: All comments have been addressed

2. Is the manuscript technically sound, and do the data support the conclusions?

Reviewer #1: Yes

Reviewer #2: Partly

3. Has the statistical analysis been performed appropriately and rigorously? 

Reviewer #1: Yes

Reviewer #2: No

4. Have the authors made all data underlying the findings in their manuscript fully available?

Reviewer #1: Yes

Reviewer #2: Yes

5. Is the manuscript presented in an intelligible fashion and written in standard English?

Reviewer #1: Yes

Reviewer #2: No

6. Review Comments to the Author

Reviewer #1: All changes made and the paper is stronger for it. Thank you for your hard work, and well done. Accept.

Reviewer #2: Thank you for the opportunity to review the revised manuscript entitled The consumption of alcohol by adolescent schoolchildren: Differences in the triadic relationship pattern between rural and urban environments submitted to PLOS One (PONE-D-20-13204R1). The manuscript has improved but there remains significant problems, mainly related to the language and grammar. Specific comments are below.

The entire abstract needs copyediting. Adolescence is confused with adolescents. Analyse should be analyze. The methodology section should be expanded. Results from the regression models should be added to the abstract.

Similarly, the introduction needs significant copyediting.

The sample population and recruitment should start the methods section. Was assent obtained for minors? What does the initial population refer to (students that expressed interest)? What geographic unit/size for classifying urban/rural, mentioning 50,000 population alone is insufficient. Nuclei? A better description of access between rural and urban areas would be appreciated. The target population described at the end of the first paragraph on page 6 should be moved to the beginning of the methods section and this is unclear. The results of the power analysis should be mentioned in the text. The analyses should be reviewed by a statistician. The entire methods section needs copyediting.

The study overall is not as strong as it could be. The sample size is relatively small and the analyses are not well described.

7. PLOS authors have the option to publish the peer review history of their article (what does this mean?). If published, this will include your full peer review and any attached files.

Reviewer #1: No

Reviewer #2: No

---

## [Author Response · Author response to Decision Letter 1]

16 Sep 2020

Reviewer 1

Reviewer #1: All changes made and the paper is stronger for it. Thank you for your hard work, and well done. Accept.

Answer to reviewer #1: We appreciate the review and are pleased to know that all suggestions have been properly addressed.

Reviewer 2

Reviewer #2: Thank you for the opportunity to review the revised manuscript entitled The consumption of alcohol by adolescent schoolchildren: Differences in the triadic relationship pattern between rural and urban environments submitted to PLOS One (PONE-D-20-13204R1). The manuscript has improved but there remains significant problems, mainly related to the language and grammar. Specific comments are below. 

Answer to R2 0: We appreciate your review and are pleased to learn that the manuscript has improved following the changes made. We hope to be able to satisfy the new comments indicated in this revision.

R2.1:

The entire abstract needs copyediting. Adolescence is confused with adolescents. Analyse should be analyze. The methodology section should be expanded. Results from the regression models should be added to the abstract. 

Answer to R2.1:

Taking into account your suggestions, we have hired a translator who has been responsible for the copyediting of the entire manuscript.

In the manuscript "with tracked changes" it is possible to see the modifications made by the translator.

In addition, we have modified the abstract by adding more information, as suggested.

R2.2:

Similarly, the introduction needs significant copyediting. 

Answer to R2.2:

The translator performed this task. We hope that the section will be better written.

R2.3:

The sample population and recruitment should start the methods section. Was assent obtained for minors? 

Answer to R2.3:

Following this suggestion, we have added this information to the manuscript in the first paragraph of section 2.

R2.4:

What does the initial population refer to (students that expressed interest)? What geographic unit/size for classifying urban/rural, mentioning 50,000 population alone is insufficient. Nuclei? A better description of access between rural and urban areas would be appreciated. 

Answer to R2.4:

In response to this suggestion, we have added more justified information with scientific references.

“The total population of the different institutes was 270 adolescents. Of these, 195 adolescents or their parents in the case of adolescents under 18 years of age signed the informed consent. 

In this way, the sample was made up of 195 adolescents aged between 16 and 19, attending rural and urban public schools in the area of El Bierzo (León, Spain), during the 2015-2016 academic year. The selection was with the consent of the project from the centers. We also took into account the relationships of the research team with the directors and professors, which facilitated the launch of the research. A search for educational centers in rural and urban settings was carried out in order to obtain a sample of similar size, which would allow us to compare both rural and urban areas.

There are various definitions for characterising rural settings. The Ministry of the Environment defines the proportion and density of the population as the most used criterion. In Spain, rural areas are defined as "A geographic space comprising the aggregation of municipalities or smaller local entities with a population <30,000 inhabitants and a density <100 inhabitants per Km 2 [26].”

R2.5:

The target population described at the end of the first paragraph on page 6 should be moved to the beginning of the methods section and this is unclear. 

Answer to R2.5:

In line with this suggestion, the text has been amended and moved to the position indicated.

R2.6:

The results of the power analysis should be mentioned in the text. The analyses should be reviewed by a statistician. The entire methods section needs copyediting. 

Answer to R2.6:

We have included a comment about the power analysis in the text. The results of the statistical analysis have been revised and reported in the text as indicated by the statistician.

R2.7:

The study overall is not as strong as it could be. The sample size is relatively small and the analyses are not well described.

Answer to R2.7:

We hope that after all the proposed changes that have been made, we will have fulfilled this requirement as well.

---

## [Decision Letter · Decision Letter 2]

5 Oct 2020

PONE-D-20-13204R2

The consumption of alcohol by adolescent schoolchildren: Differences in the triadic relationship pattern between rural and urban environments.

PLOS ONE

Dear Dr. Quiroga Sánchez,

Thank you for submitting your manuscript to PLOS ONE. After careful consideration, we feel that it has merit but does not fully meet PLOS ONE’s publication criteria as it currently stands. Therefore, we invite you to submit a revised version of the manuscript that addresses the points raised during the review process.

We look forward to receiving your revised manuscript.

Kind regards,

Joel Msafiri Francis, MD, MS, PhD

Academic Editor

PLOS ONE

Additional Editor Comments (if provided):

The manuscript will benefit from copy editing. It would be helpful to seek help from the native English speaker.

Reviewers' comments:

Reviewer's Responses to Questions

**Comments to the Author**

1. If the authors have adequately addressed your comments raised in a previous round of review and you feel that this manuscript is now acceptable for publication, you may indicate that here to bypass the “Comments to the Author” section, enter your conflict of interest statement in the “Confidential to Editor” section, and submit your "Accept" recommendation.

Reviewer #1: All comments have been addressed

Reviewer #2: (No Response)

2. Is the manuscript technically sound, and do the data support the conclusions?

Reviewer #1: Yes

Reviewer #2: (No Response)

3. Has the statistical analysis been performed appropriately and rigorously? 

Reviewer #1: Yes

Reviewer #2: (No Response)

4. Have the authors made all data underlying the findings in their manuscript fully available?

Reviewer #1: No

Reviewer #2: (No Response)

5. Is the manuscript presented in an intelligible fashion and written in standard English?

Reviewer #1: Yes

Reviewer #2: (No Response)

6. Review Comments to the Author

Reviewer #1: I had not requested any further comments so all good on my side and wishing you the best of luck with your paper.

Reviewer #2: The authors have addressed the comments. Grammar and readability continues to be a problem. Methodology/analyses appropriate.

7. PLOS authors have the option to publish the peer review history of their article (what does this mean?). If published, this will include your full peer review and any attached files.

Reviewer #1: No

Reviewer #2: No

---

## [Author Response · Author response to Decision Letter 2]

7 Oct 2020

Additional Editor Comments (if provided):

The manuscript will benefit from copy editing. It would be helpful to seek help from the native English speaker.

Answer to AE:

First of all, I would like to thank the reviewers for their work. We are pleased to have solved all the suggestions indicated in the two reviews by the two reviewers.

Following the suggestions of the reviewers and the editor, we have sent our article to the company American Journal Experts (AJE) for copyediting. I have attached the certificate of review with the code TL4M4QZ6 in the inventory of files of the journal.

Reviewer 1

Reviewer #1: I had not requested any further comments so all good on my side and wishing you the best of luck with your paper.

Answer to R1:

We are glad to know that we have taken care of all your suggestions correctly. Thank you very much for your message.

Reviewer 2

Reviewer #2: The authors have addressed the comments. Grammar and readability continues to be a problem. Methodology/analyses appropriate.

Answer to R2:

We are glad to know that we have taken care of all your suggestions correctly. Thank you very much for your message.

Following your suggestions, we have sent our manuscript to the company American Journal Experts (AJE) for copyediting.

---

## [Editor Report · Decision Letter 3]

9 Oct 2020

The consumption of alcohol by adolescent schoolchildren: Differences in the triadic relationship pattern between rural and urban environments.

PONE-D-20-13204R3

Dear Dr. Quiroga Sánchez,

We’re pleased to inform you that your manuscript has been judged scientifically suitable for publication and will be formally accepted for publication once it meets all outstanding technical requirements.

Kind regards,

Joel Msafiri Francis, MD, MS, PhD

Academic Editor

PLOS ONE
---

## [Editor Report · Acceptance letter]

13 Oct 2020

PONE-D-20-13204R3 

The consumption of alcohol by adolescent schoolchildren: Differences in the triadic relationship pattern between rural and urban environments. 

Dear Dr. Quiroga Sánchez:

I'm pleased to inform you that your manuscript has been deemed suitable for publication in PLOS ONE. Congratulations! Your manuscript is now with our production department. 

Kind regards, 

on behalf of

Dr. Joel Msafiri Francis 

Academic Editor

PLOS ONE